# Three New *Periconia* Species Isolated from *Wurfbainia villosa* in Guangdong, China: A Discussion on the Doubtful Taxa Clustering in this Genus

Chunfang Liao [1,2,3], Kevin D. Hyde [1,2,3], Kandawatte Wedaralalage Thilini Chethana [1,2,3], Wei Dong [1], Yunhui Yang [1,2,3] and Mingkwan Doilom [1,*]

1   Innovative Institute for Plant Health/Key Laboratory of Green Prevention and Control on Fruits and Vegetables in South China, Ministry of Agriculture and Rural Affairs, Zhongkai University of Agriculture and Engineering, Guangzhou 510225, China; 6371105002@lamduan.mfu.ac.th (C.L.); kdhyde3@gmail.com (K.D.H.); kandawatte.thi@mfu.ac.th (K.W.T.C.); dongwei0312@hotmail.com (W.D.); yyunhui1226@163.com (Y.Y.)
2   Center of Excellence in Fungal Research, Mae Fah Luang University, Chiang Rai 57100, Thailand
3   School of Science, Mae Fah Luang University, Chiang Rai 57100, Thailand
*   Correspondence: j_hammochi@hotmail.com; Tel.: +86-15902010761

**Abstract:** During a survey of fungi on *Wurfbainia villosa* in Guangdong Province, China, three novel species, *Periconia endophytica*, *P. yangjiangensis*, and *P. wurfbainiae*, belonging to Periconiaceae in Pleosporales, Dothideomycetes are proposed based on morphological and phylogenetic evidence. *Periconia endophytica* was isolated from the healthy leaves of *W. villosa*, while *P. yangjiangensis* and *P. wurfbainiae* were obtained from the dead stems of the same host. Notably, holomorphs were observed in *P. wurfbainiae*. The morphological characteristics of the novel taxa are compared with closely related species within *Periconia*. Illustrations, morphological descriptions, and phylogenetic analyses are provided for the novel taxa. Multilocus phylogeny of the combined internal transcribed spacer (ITS), large subunit nuclear rDNA (LSU), small subunit nuclear ribosomal rDNA (SSU), and partial translation elongation factor 1–α (*tef1-α*) regions supported the establishment of three new species. Furthermore, the taxa clustering in *Periconia*, *Flavomyces fulophazii*, and *Sporidesmium tengii*, are discussed for further investigation of their taxonomic placements.

**Keywords:** three new species; traditional medicinal plant; microfungi; Periconiaceae; Ascomycota; Zingiberaceae; multilocus phylogeny; taxonomy

## 1. Introduction

*Wurfbainia villosa* is a perennial, evergreen herb belonging to the ginger family (Zingiberaceae: Alpinioideae). This traditional medicinal herb is one of the four major southern Chinese medicines [1]. The fruits of *W. villosa* are commonly used in traditional Chinese medicine due to their numerous therapeutic properties, including dampness elimination, diarrhea treatment, miscarriage prevention, spleen warming, and as appetite stimulants [2]. *Wurfbainia villosa* is mainly distributed in southeast and east Asia (e.g., Cambodia, China, India, Indonesia, Laos, Myanmar, Thailand, and Vietnam) [3,4]. In China, it is distributed in Fujian, Guangdong, Guangxi, and Yunnan provinces [5]. Yangchun City in the Guangdong Province is widely recognized as a hub for sourcing this medicinal herb [6]. However, limited taxonomic and phylogenetic information is available regarding the fungi associated with *W. villosa*, particularly in Yangchun City. A total of six fungal species have been reported on this medicinal plant in this region [7,8]. During a survey conducted in Yangchun City, Guangdong Province, China, to investigate microfungi associated with the traditional medicinal plant *W. villosa*, ascomycetous fungi resembling *Periconia* were isolated from the dead stems and healthy leaves of this plant.

*Periconia* was introduced by Tode [9] with *P. lichenoides* as the type species. *Periconia* was previously classified under Massarinaceae [10]. Subsequently, Tanaka et al. [11] reclassified *Periconia* in Periconiaceae based on the multi-locus phylogenetic analysis, which showed that *Periconia* species formed a distinct cluster separate from Massarinaceae. Until now, the latest treatment of *Periconia* by Tanaka et al. [11] has been followed by subsequent studies [12–17]. Most *Periconia* species have been reported based on their asexual morphs, while five species have been reported based on their sexual morphs, viz., *P. didymosporum*, *P. homothallica*, *P. igniaria*, *P. prolifica*, and *P. pseudodigitata* [11,17]. The asexual morph is mainly characterized by macronematous, mononematous, brown, branched, or unbranched conidiophores, with brown, ovoid to clavate or spherical conidial heads, intercalary or terminal conidiogenous cells with catenate or solitary, golden brown to dark brown, globose to subglobose, aseptate, smooth or verruculose conidia [11,15,18–21]. The sexual morph is characterized by immersed or erumpent, scattered or aggregated, globose to subglobose, ostiolate ascomata, hyaline periphyses, 8-spored, bitunicate, cylindrical, pedicellate asci, with an ocular chamber, and fusiform, 1-septate, hyaline, guttulate, smooth-walled ascospores with an entire sheath [16,17,22,23].

*Periconia* species have been reported as saprobes, endophytes, plant and human pathogens, distributed widely in terrestrial habitats and rarely in aquatic and marine environments [24–27]. *Periconia byssoides* was isolated from the gastrointestinal tract of sea hares (*Aplysia kurodai*) [28]. *Periconia aquatica* and *P. submersa* were isolated from decaying wood in the Nujiang River [12]. *Periconia circinata* was isolated as a pathogen causing blackening and rotting of wheat roots and stem bases [24], while *P. igniaria* and *P. macrospinosa* were reported as pathogens causing leaf spots on yellow starthistle [29] and leaf necrosis of pointed gourd [30], respectively. Furthermore, *P. keratitis* was reported as a human pathogen causing mycotic keratitis in India [27].

*Periconia* is the source of many economically important bioactive compounds [31]. From 1969 to the present, more than 100 compounds have been isolated from *Periconia* species, including aromatic compounds, carbohydrate derivatives, cytochalasans, macrolides, macropshelides, meroterpenes, polyketides, and terpenoids [25,28,31–35]. *Periconia atropurpurea* isolated from the leaves of *Xylopia aromatica* was reported to produce aromatic compounds [25]. Moreover, *Periconia* sp., isolated from branches of *Taxus cuspidata*, produces two fusicoccane diterpenes, namely periconicins A (1) and B (2), exhibiting potent antibacterial activity [36]. Therefore, a taxonomic study of the *Periconia* fungi could facilitate species identification to explore their potential for bioactive compounds.

In this study, we have reported three new species, *Periconia endophytica*, *P. yangjiangensis*, and *P. wurfbainiae*, based on morphological and multilocus phylogenetic analyses. The saprobic taxa, *P. yangjiangensis* and *P. wurfbainiae*, as well as the endophytic taxon, *P. endophytica*, were respectively isolated from decaying stems and healthy leaves of *Wurfbainia villosa* in Guangdong Province, China. Both asexual and sexual morphs are described for *P. wurfbainiae*. Furthermore, we also discussed the clustering of doubtful taxa, namely *Flavomyces fulophazii* and *Sporidesmium tengii*, within the genus *Periconia*.

## 2. Materials and Methods

### 2.1. Collection, Observation and Isolation of Samples

Endophytic fungi were isolated from the healthy leaves of *Wurfbainia villosa* collected in Yongning Town, Yangchun City, Guangdong Province, China in 2022. Fresh and healthy leaves were washed under tap water and dried with tissue paper, then cut into 0.3–0.5 cm × 0.3–0.5 cm sections using scissors. The leaf fragments were surface sterilized in 2.5% sodium hypochlorite for 1 min, 75% ethanol for 2 min, washed three times in sterilized water for 3 min, and finally dried on tissue filter paper. The surface-sterilized plant fragments were aseptically transferred onto PDA containing chloramphenicol (30 mg/mL) and incubated at 25 °C. The hyphae were purified two to three times under sterile conditions using a SZ650 stereo microscope in order to obtain a pure culture.

Saprobic samples were collected from the dead stem of *Wurfbainia villosa* in Heshui Town, Yangchun City, Guangdong Province, China in 2022. The methods used for capturing the observed samples and fungal structures, such as ascomata, hamathecium, asci, ascospores, conidiophores, conidiogenous cells, and conidia followed the protocol described by Liao et al. [8]. The isolation of single spores was conducted according to the methods described by Senanayake et al. [37].

The fungal isolates were preserved, fungal structures were measured, and herbarium specimens and living cultures were deposited according to the protocol described by Liao et al. [8]. The new species in this study were registered in Faces of Fungi (FoF) (http://www.facesoffungi.org, accessed on 30 December 2023) and Index Fungorum (IF) databases (http://www.indexfungorum.org/names/names.asp, accessed on 30 December 2023).

### 2.2. DNA Extraction, PCR Amplification, and Sequencing

Fungal genomic DNA was extracted using the methods described by Liao et al. [8], with adherence to the manufacturer's instructions (Guangzhou Magen Biotechnology Co., Ltd., Guangzhou, China). Extracted DNA was stored at $-20\ ^\circ$C for future PCR amplification. The internal transcribed spacer (ITS), large subunit rDNA (LSU), small subunit nuclear ribosomal rDNA (SSU), and partial translation elongation factor 1–$\alpha$ (*tef1-$\alpha$*) were amplified and sequenced using primers ITS1 and ITS4 [38], LR5 and LR0R [39], NS1 and NS4 [38], and EF1-728F and EF2 [40], respectively.

The polymerase chain reaction (PCR) was performed with a volume of 25 μL, and the thermal cycling program was used to amplify ITS, LSU, and *tef1-$\alpha$* following the protocol described by Liao et al. [8]. SSU amplification was conducted according to the ITS protocol. PCR products were purified and sequenced by Tianyi Huiyuan Gene Technology & Services Co. (Guangzhou, China). The sequences in this study were submitted to GenBank (Table 1).

**Table 1.** The GenBank accession numbers of the taxa utilized in the phylogenetic analyses conducted in this study.

| Species Name | Strain Number | GenBank Accession Number | | | |
| --- | --- | --- | --- | --- | --- |
| | | ITS | LSU | SSU | *tef1-α* |
| *Flavomyces fulophazae* | CBS 135664 | KP184000 | KP184039 | KP184081 | – |
| *F. fulophazae* [T] | CBS 135761 | NR_137960 | NG_058131 | NG_061191 | – |
| *Lentithecium aquaticum* [T] | CBS 123099 | NR_160229 | NG_064211 | NG_016507 | GU349068 |
| *L. clioninum* [T] | KT 1149A | LC014566 | AB807540 | AB797250 | AB808515 |
| *L. clioninum* | KT 1220 | LC014567 | AB807541 | AB797251 | AB808516 |
| *Massarina cisti* [T] | CBS 266.62 | – | AB807539 | AB797249 | AB808514 |
| *M. eburnea* | CBS 473.64 | – | GU301840 | GU296170 | GU349040 |
| *Morosphaeria ramunculicola* | KH 220 | – | AB807554 | AB797264 | AB808530 |
| *M. velatispora* | KH 221 | LC014572 | AB807556 | AB797266 | AB808532 |
| *Periconia alishanica* | KUMCC 19-0174 | MW063167 | MW063231 | – | MW183792 |
| *P. alishanica* [T] | MFLUCC 19-0145 | MW063165 | MW063229 | – | MW183790 |
| *P. ananasi* [T] | MFLUCC 21-0155 | OL753685 | OL606153 | OL606142 | OL912946 |
| *P. ananasi* | KUMCC 21-0470 | OM102539 | OL985955 | OL979226 | OM007977 |
| *P. aquatica* [T] | MFLUCC 16-0912 | KY794701 | KY794705 | – | KY814760 |
| *P. artemisiae* [T] | KUMCC 20-0265 | MW448657 | MW448571 | MW448658 | MW460898 |
| *P. banksiae* [T] | CBS 129526 | – | NG_064279 | – | – |
| *P. byssoides* | KUMCC 20-0264 | MW444854 | MW444855 | MW444856 | MW460895 |
| *P. byssoides* | MFLUCC 17-2292 | MK347751 | MK347968 | MK347858 | MK360069 |
| *P. byssoides* | MFLUCC 18-1548 | MK347794 | MK348013 | MK347902 | MK360070 |
| *P. byssoides* | MFLUCC 18-1553 | MK347806 | MK348025 | MK347914 | MK360068 |
| *P. byssoides* [T] | MFLUCC 20-0172 | MW063162 | MW063226 | – | – |
| *P. byssoides* | NCYUCC 19-0314 | MW063163 | MW063227 | – | – |
| *P. caespitosa* [T] | LAMIC 110 16 | MH051906 | MH051907 | – | – |
| *P. chengduensis* [T] | CGMCC 3.23930 | OP955987 | OP956012 | OP956056 | OP961453 |

**Table 1.** *Cont.*

| Species Name | Strain Number | GenBank Accession Number | | | |
|---|---|---|---|---|---|
| | | ITS | LSU | SSU | *tef1-α* |
| *P. chengduensis* | UESTCC 22.0140 | OP955977 | OP956002 | OP956046 | OP961443 |
| *P. chengduensis* | UESTCC 22.0142 | OP955978 | OP956003 | OP956047 | OP961444 |
| *P. chimonanthi* [T] | KUMCC 20-0266 | NR_176752 | MW448572 | MW448656 | MW460897 |
| *P. chimonanthi* | UESTCC 22.0133 | OP955964 | OP955989 | OP956033 | OP961430 |
| *P. citlaltepetlensis* [T] | IOM 325319 | MH890645 | MT625978 | – | – |
| *P. citlaltepetlensis* | IOM 325319.2 | MT649221 | MT649216 | – | – |
| *P. cookei* | MFLUCC 17-1399 | MG333490 | MG333493 | – | MG438279 |
| *P. cookei* | MFLUCC 17-1679 | – | MG333492 | – | MG438278 |
| *P. cortaderiae* | MFLUCC 15-0451 | KX965734 | KX954403 | KX986346 | KY429208 |
| *P. cortaderiae* [T] | MFLUCC 15-0457 | KX965732 | KX954401 | KX986345 | KY310703 |
| *P. cynodontis* [T] | CGMCC 3.23927 | OP909925 | OP909921 | OP909920 | OP961434 |
| *P. cyperacearum* [T] | CPC 32138 | NR_160357 | NG_064549 | – | – |
| *P. delonicis* [T] | MFLUCC 17-2584 | – | NG_068611 | NG_065770 | MK360071 |
| *P. didymosporum* [T] | MFLU 15-0058 | KP761734 | KP761731 | KP761738 | KP761728 |
| *P. digitata* | CBS 510.77 | LC014584 | AB807561 | AB797271 | AB808537 |
| *P. elaeidis* [T] | MFLUCC 17-0087 | MG742713 | MH108552 | MH108551 | – |
| *P. endophytica* [T] | ZHKUCC 23-0995 | OR995582 | OR995588 | PP277722 | PP025968 |
| *P. endophytica* | ZHKUCC 23-0996 | OR995583 | OR995589 | PP277723 | PP025969 |
| *P. epilithographicola* [T] | CBS 144017 | NR_157477 | – | – | – |
| *P. epilithographicola* [T] | MFLUCC 21-0153 | OL753687 | OL606155 | OL606144 | OL912948 |
| *P. festucae* [T] | CGMCC 3.23929 | OP955973 | OP955998 | OP956042 | OP961439 |
| *P. philadelphiana* [T] | CPC 42854 | OQ628486 | OQ629068 | – | – |
| *P. homothallica* [T] | KT 916 | AB809645 | AB807565 | AB797275 | |
| *P. igniaria* | CBS 845.96 | LC014586 | AB807567 | AB797277 | AB808543 |
| *P. imperatae* [T] | CGMCC 3.23931 = UESTCC 22.0129 | OP955984 | OP956009 | OP956053 | OP961450 |
| *P. imperatae* | UESTCC 22.0145 | OP955979 | OP956004 | OP956048 | OP961445 |
| *P. imperatae* | UESTCC 22.0146 | OP955983 | OP956008 | OP956052 | OP961449 |
| *P. macrospinosa* | CBS 135663 | KP183999 | KP184038 | KP184080 | – |
| *P. minutissima* | MFLUCC 15-0245 | KY794703 | KY794707 | – | – |
| *P. minutissima* | MUT 2887 | MG813227 | – | – | – |
| *P. neobrittanica* [T] | CPC 37903 | NR_166344 | NG_068342 | – | – |
| *P. neominutissima* [T] | CPC 42368 | OQ628478 | OQ629060 | – | – |
| *P. palmicola* [T] | MFLUCC 14-0400 | – | NG_068917 | MN648319 | MN821070 |
| *P. penniseti* [T] | CGMCC 3.23928 | OP955971 | OP955996 | OP956040 | OP961437 |
| *P. prolifica* | DBOF74 | JQ724435 | – | – | – |
| *P. prolifica* | DBOF129 | JQ724490 | – | – | – |
| *P. pseudobyssoides* | DUCC 0850 | MG333491 | MG333494 | – | MG438280 |
| *P. pseudobyssoides* | KUMCC 20-0263 | MW444851 | MW444852 | MW444853 | MW460894 |
| *P. pseudodigitata* | KT 644 | LC014589 | AB807562 | AB797272 | AB808538 |
| *P. pseudodigitata* [T] | KT 1395 | NR_153490 | NG_059396 | NG_064850 | AB808540 |
| *P. spodiopogonis* | CGMCC 3.23932 | OP955963 | OP955988 | OP956032 | OP961429 |
| *P. salina* [T] | MFLU 19-1235 | MN047086 | MN017846 | MN017912 | – |
| *P. submersa* [T] | MFLUCC 16-1098 | KY794702 | KY794706 | – | KY814761 |
| *P. thailandica* [T] | MFLUCC 17-0065 | KY753887 | KY753888 | KY753889 | – |
| *P. thysanolaenae* [T] | KUMCC 20-0262 | MW442967 | MW444850 | MW448659 | MW460896 |
| *P. variicolor* [T] | SACCR-64 | DQ336713 | – | – | – |
| *P. verrucosa* [T] | MFLUCC 17-2158 | MT310617 | MT214572 | MT226686 | MT394631 |
| *P. verrucosa* | UESTCC 22.0149 | OP955976 | OP956001 | OP956045 | OP961442 |
| *P. verrucosa* | UESTCC 22.0150 | OP955980 | OP956005 | OP956049 | OP961446 |
| *P. wurfbainiae* [T] | ZHKUCC 23-0999 | OR995586 | OR995592 | PP277726 | PP025972 |
| *P. wurfbainiae* | ZHKUCC 23-1000 | OR995587 | OR995593 | PP277727 | PP025973 |
| *P. yangjiangensis* [T] | ZHKUCC 23-0997 | OR995584 | OR995590 | PP277724 | PP025970 |
| *P. yangjiangensis* | ZHKUCC 23-0998 | OR995585 | OR995591 | PP277725 | PP025971 |
| *Sporidesmium tengii* | HKUCC 10837 | – | DQ408559 | – | – |

Notes: The newly generated sequences in this study are shown in blue, while the ex-type strains are denoted by "T". Abbreviations: CBS: Culture Collection of the Westerdijk Fungal Biodiversity Institute, Utrecht, Netherlands; CGMCC: China General Microbiological Culture Collection Center Beijing, China; CPC: Culture Collection of Pedro Crous, Netherlands; DBOF: DNA Barcoding Ocean Fungi; DUCC: Dali University Culture Collection, Yunnan, China; IOM: Instituto de Oftalmología "Fundación Conde de Valenciana" IAP Mexico Culture Collection; KH: K. Hirayama; KT: Kaz. Tanaka; KUMCC: Kunming Institute of Botany Culture Collection, Kunming, China; LAMIC: Laboratorio de Micologia, Universidade Estadual de Feira de Santana, Brazil; MFLU: Herbarium of Mae Fah Luang University, Chiang Rai, Thailand; MFLUCC: Mae Fah Luang University Culture Collection, Chiang Rai, Thailand; MUT: Mycotheca Universitatis Taurinensis, Department of Life Sciences and Systems Biology, University of Turin, Turin, Italy; NCYUCC: National Chiayi University Culture Collection, Taiwan, China; UESTCC: University of Electronic Science and Technology Culture Collection, Chengdu, China.

*2.3. Phylogenetic Analyses*

The sequences used for phylogenetic analyses were downloaded from GenBank and corresponded to published literature based on the Blastn search of the ITS gene against the GenBank database [16,17]. A total of 78 sequences were used in the phylogenetic analyses (Table 1), with *Morosphaeria ramunculicola* (KH 220) and *M. velatispora* (KH 221) as the outgroup taxa. Four single-gene loci were aligned in the MAFFT v. 7 online program. The sequence was trimmed with Trimal version v. 3 (Gappyout option) [41]. Alignments were converted to NEXUS format using the Alignment Transformation Environment online platform (http://www.sing-group.org/ALTER/, accessed on 30 December 2023).

Phylogenies based on the combined ITS, LSU, SSU, and *tef1-α* sequence data were generated using maximum-likelihood (ML) and Bayesian inference (BI) analyses performed in the CIPRES Science Gateway online platform (https://www.phylo.org/portal2/home.action, accessed on 30 December 2023). The maximum likelihood (ML) analysis was conducted via RA × ML-HPC2 on XSEDE (8.2.10), with the GTR + G + I evolutionary substitution model and 1000 replicates of rapid bootstrap inferences. The free model parameters were estimated using RA × ML ML with 25 per site rate categories. The likelihood of the final tree was evaluated and optimized using GAMMA. The jModelTest2 on XSEDE (2.1.6) online platform (https://www.phylo.org/portal2/createTask.action, accessed on 30 December 2023) was performed for each locus to estimate the best-fit evolutionary model under the Akaike Information Criterion (AIC) [42]. The Bayesian inference (BI) analysis was conducted with the Markov Chain Monte Carlo (MCMC) method in MrBayes version 3.2.7a on the XSEDE tool on the CIPRES portal [43]. Four simultaneous Markov chains were run for 5,000,000 generations. The phylogenetic trees were sampled every 100th generation. The resulting trees were visualized in FigTree v. 1.4.0 [44] and formatted using PowerPoint 2010 (Microsoft Corporation, Redmond, WA, USA).

## 3. Results

*3.1. Phylogeny*

The phylogenetic tree of the combined ITS (1–605 bp), LSU (606–1450), SSU (1451–2470), and *tef1-α* (2471–3403) sequence data comprised 78 taxa, including *Morosphaeria ramunculicola* (KH 220) and *M. velatispora* (KH 221) as the outgroup taxa. The topology of the ML analysis was similar to the BI analysis, and the best ML tree with a final ML optimization likelihood value of −19495.411175 is shown in Figure 1. The matrix, comprising a total of 3403 characters, exhibits 1235 distinct alignment patterns, with 32.42% undetermined characters or gaps. Estimated base frequencies were as follows: A = 0.236736, C = 0.256383, G = 0.268919, T = 0.237962; substation rates: AC = 1.512523, AG = 2.690423, AT = 1.737420, CG = 1.087709, CT = 9.882829, GT = 1.000000; gamma distribution shape parameter: $\alpha = 0.206248$.

In this study, our tree topology is similar to the previous studies of Yang et al. [16] and Su et al. [17]. Nonetheless, the inclusion of newly discovered taxa, such as the six novel species (*P. chengduensis*, *P. cynodontis*, *P. festucae*, *P. imperatae*, *P. penniseti*, and *P. spodiopogonis*) reported by Su et al. [17], has led to minor changes in the positions of certain species. The addition of our novel species resulted in *Sporidesmium tengii* (HKUCC 10837) clustering with *Periconia cynodontis* CGMCC 3.23927 instead of *P. didymosporium* MFLU 15-0058 as in Yang et al. [16]. Phylogenetic analyses showed that our six strains clustered within *Periconia*. Two strains, ZHKUCC 23-0997 and ZHKUCC 23-0998 of *P. yangjiangensis*, formed a distinct lineage, separated from other *Periconia* species with 95% ML and 1.00 BYPP. Two strains of *P. endophytica*, ZHKUCC 23-0995 and ZHKUCC 23-0996, formed a distinct branch, separated from other *Periconia* species with 89% ML and 1.00 BYPP. Two *P. wurfbainiae* strains, ZHKUCC 23-0999 and ZHKUCC 23-1000, formed a distinct branch, separated from other *Periconia* species with 62% ML and 0.94 BYPP (Figure 1).

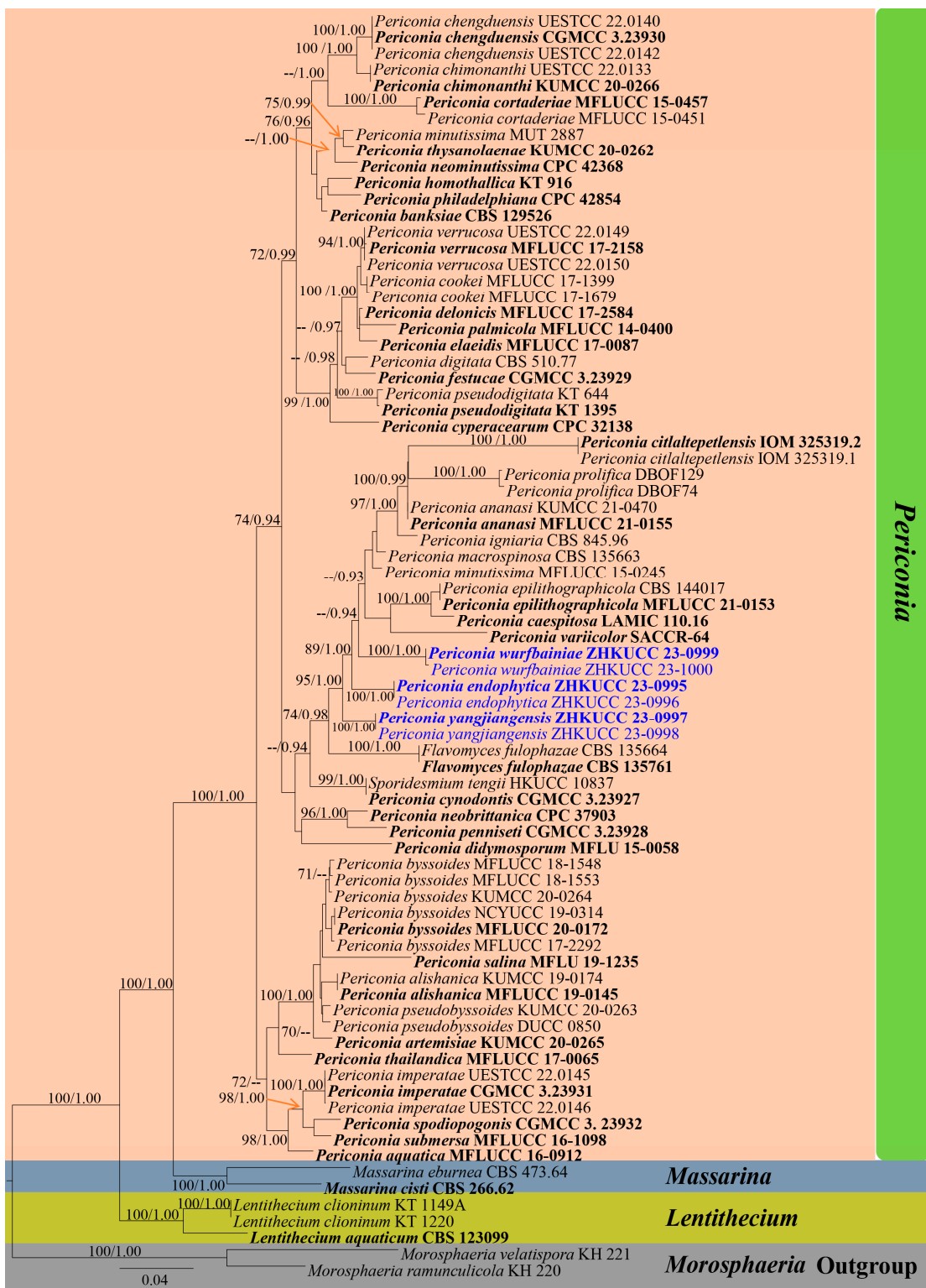

**Figure 1.** The maximum likelihood (ML) tree is based on the combined ITS, LSU, SSU, and *tef1-α* sequence data. Bootstrap support values with an ML greater than 70% and Bayesian posterior probabilities (PP) greater than 0.90 are given above the nodes, shown as "ML/PP". The tree is rooted with *Morosphaeria ramunculicola* (KH 220) and *M. velatispora* (KH 221). Isolates of the current study are indicated in blue and type strains are in bold. The different colors of the background indicated different genera, with color highlighting *Periconia*. The arrows indicate ML and PP support for each node when space is limited.

### 3.2. Taxonomy

*Periconia endophytica* C.F. Liao, Doilom & K.D. Hyde, sp. nov.
Index Fungorum number: IF851407; Facesoffungi number: FoF 15276; Figure 2.

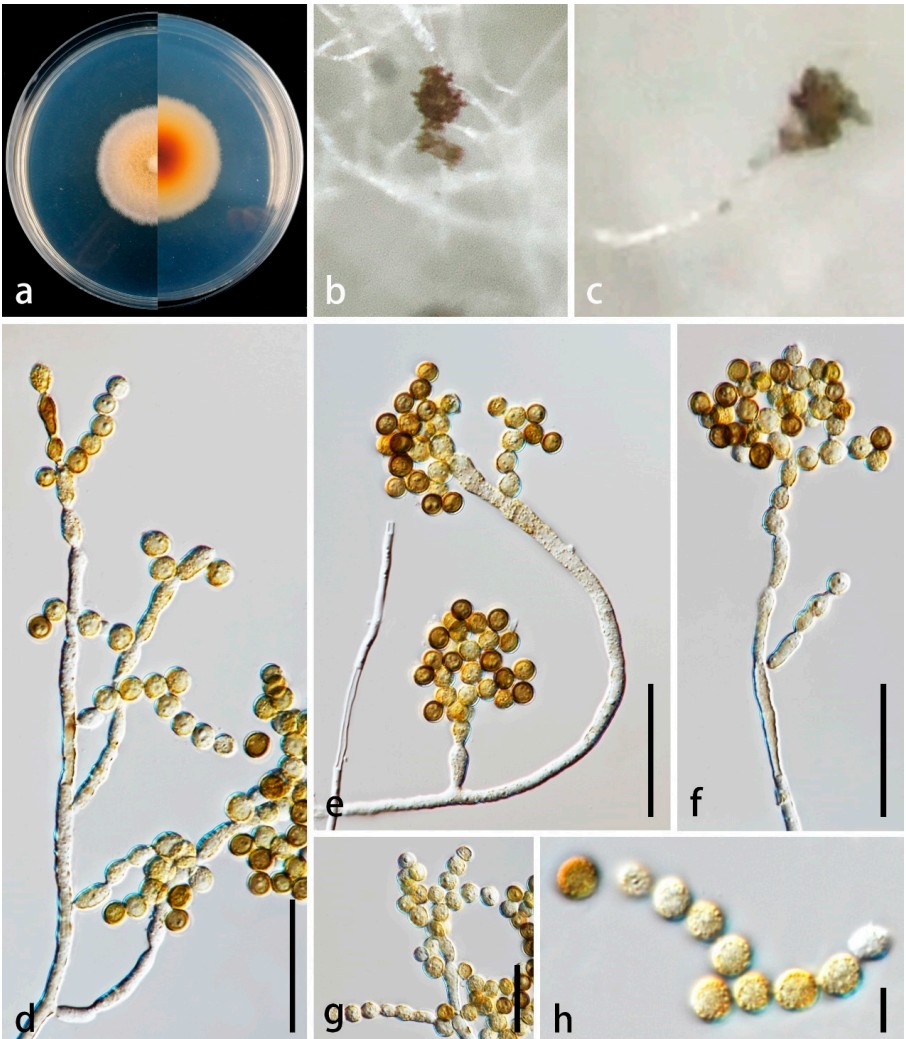

**Figure 2.** *Periconia endophytica* (MHZU 23-0245, holotype). (**a**) Upper view and reverse view of culture on PDA. (**b**,**c**) Close-up of mycelia and conidia on the sterile plant tissue in PDA after a month of incubation. (**d**–**g**) Conidiophores, conidiogenous cells with conidia. (**h**) Conidia. Scale bars: (**d**–**f**) = 50 μm, (**g**) = 20 μm, (**h**) = 5 μm.

Etymology. The epithet "endophytica" refers to the endophytic lifestyle of the species.
Holotype. MHZU 23-0245

Endophytic associated with leaves of *Wurfbainia villosa*. Sexual morph: undetermined. Asexual morph: sporulation on the sterile plant tissue in PDA after a month of incubation, spore masses visible as black spot, scattered on colonies. Hyphae 1.5–3.5 μm wide ($\bar{x}$ = 2.5 μm, *n* = 30), branched, hyaline, septate, smooth to slightly verruculose. Conidiophores 15–110 × 2.5–5.5 μm ($\bar{x}$ = 45 × 4 μm, *n* = 30), micro to semimacronematous, mononematous, erect, flexuous or straight, branched, sparsely septate, slightly constricted at septa, rough-walled, verruculose, thin-walled, hyaline, pale brown to yellowish brown in the upper portion. Conidiogenous cells 5–20 × 2.5–5.5 μm ($\bar{x}$ = 9 × 4 μm, *n* = 30), holoblastic, mono- and poly-blastic, integrated, determinate, terminal, subcylindrical, obovoid, ellipsoidal with truncate ends, pale brown to yellowish brown, verruculose. Conidia 4–6 × 3–5 μm ($\bar{x}$ = 5 × 4 μm, *n* = 30), globose to subglobose, aseptate, catenate, with

4–10 conidia in a chain, hyaline when young, becoming pale brown, yellowish brown and reddish brown when mature, verruculose.

Culture characteristics. Colonies on PDA reaching 4.2 cm in two weeks at $28 \pm 2$ °C, medium dense, center umbonate, circular, floccose to fluffy, velvety, filiform edge, golden brown at the center, pale brown at the margin from above; orange-brown to golden brown at the center, pale brown at the margin from reverse.

Material examined. China, Guangdong Province, Yangjiang City, Yongning Town (22.256185 N, 111.609037 E), from healthy leaves of *Wurfbainia villosa* (Zingiberaceae), 10 April 2022, C.F. Liao & Y.H. Yang, AML04C-3 (MHZU 23-0245, holotype); ex-type culture, ZHKUCC 23-0995; ibid., living culture ZHKUCC 23-0996.

Notes. Phylogenetically, *P. endophytica* formed a distinct linage and clustered with other sister *Periconia* species with 89% ML and 1.00 BYPP (Figure 1). *Periconia endophytica* differs from its phylogenetically close species *P. wurfbainiae* and *P. yangjiangensis* based on size and shape of conidiophores, conidiogenous cells, and conidia (Table 2). Based on the morphological differences and molecular support, *Periconia endophytica* is introduced as a new species.

*Periconia wurfbainiae* C.F. Liao, Doilom & K.D. Hyde, sp. nov.

Index Fungorum number: IF851409; Facesoffungi number: FoF 15277; Figures 3 and 4.

Etymology. In reference to the host genus Wurfbainia from which it was isolated.

Holotype. MHZU 23-0247

Description. Saprobic on dead stems of *Wurfbainia villosa*. Sexual morph from MHZU 23-0248: Ascomata 220–300 × 150–220 µm, solitary to scattered, immersed to semi-immersed, pyriform, black, papillate stain the substrate purple, visible as black dots on the host surface. Peridium 13–30 µm thick ($\bar{x}$ = 19 µm, *n* = 20), composed of 6–12 layers, hyaline to brown cells of *textura angularis*. Hamathecium 1.5–4 µm wide ($\bar{x}$ = 3 µm, *n* = 30), comprises of dense, trabeculate, filiform, septate, branching pseudo-paraphyses. Asci 65–100 × 9.5–15 µm ($\bar{x}$ = 81 × 12 µm, *n* = 20), 8-spored, bitunicate, cylindrical, with rounded apex and narrower, short pedicellate, with an obscured ocular chamber. Ascospores 20–26 × 5–8 µm ($\bar{x}$ = 23.5 × 7 µm, *n* = 30), overlapping uniseriate, biseriate in the middle portion, fusiform, slightly curved, 1-septate, slightly constricted at the septum, narrowly rounded at both ends, hyaline, guttulate, smooth-walled, surrounded by a mucilaginous sheath. Asexual morph from MHZU 23-0247: colonies superficial on the host substrate, hairy, solitary, gregarious, dark brown. Conidiophores 225–315 × 8–14 µm ($\bar{x}$ = 265 × 10 µm, *n* = 20), macronematous, mononematous, sometimes 1–2 cluster together on the stroma, erect, subcylindrical, with robust and swollen base, straight or slightly curved, 5–7-septate, black, unbranched, smooth-walled to slightly rough. Conidiogenous cells 7–11 × 5–9 µm ($\bar{x}$ = 8 × 6 µm, *n* = 10), holoblastic, poly-blastic, with several prominent conidiogenous loci at the apex, integrated, determinate, terminal and intercalary, reddish brown to black, subcylindrical, verruculose. Conidia 3.5–7 µm diam. ($\bar{x}$ = 5 µm, *n* = 15), globose, aseptate, catenate, usually 6–8 conidia in a chain, initially hyaline, becoming yellowish brown and greenish brown, and reddish brown to dark brown at maturity, thick-walled and verruculose.

Culture characteristics. Ascospores from specimen number MHZU 23-0248 germinating on PDA within 24h, germ tube produced from both ends. Colonies on PDA reaching 5.5 cm diam. after two weeks at room temperature ($25 \pm 2$ °C), medium dense, center umbonate, circular, center zone forming a clear bound with margin zone, floccose to fluffy, velvety, filiform edge, white from above, brown at the center, cream at the margin from reverse. Conidia from specimen number MHZU 23-0247 germinating on PDA within 12–24 h, germ tube produced from both ends. Colonies on PDA reaching 4.5 cm diam. after two weeks at room temperature ($25 \pm 2$ °C), medium dense, center umbonate, circular, floccose to fluffy, velvety, filiform edge, cream white from above, brown at the center, pale luteous at the margin from reverse. No reproduction structures of asexual morph or fruit bodies of a sexual morph on PDA.

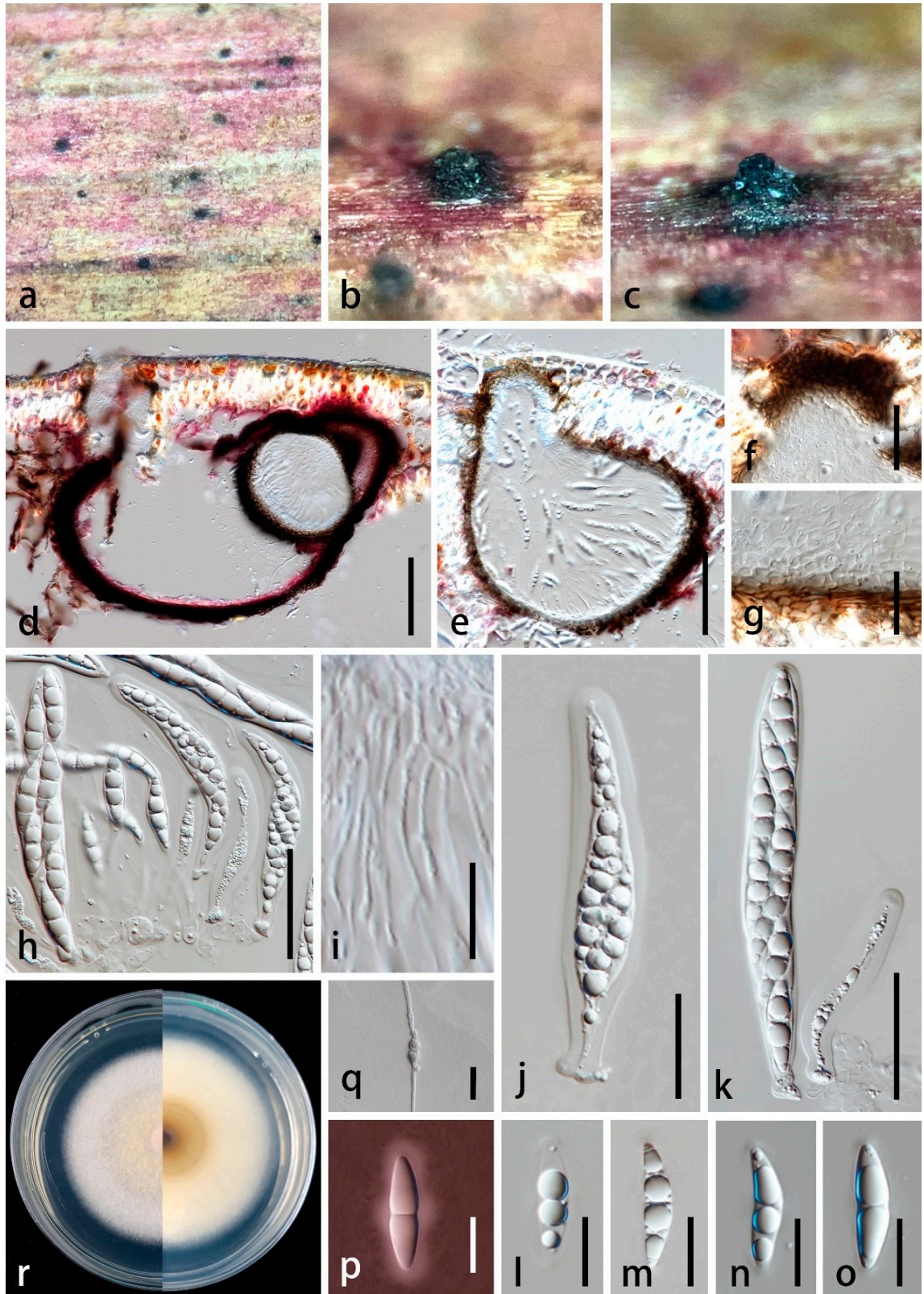

**Figure 3.** *Periconia wurfbainiae* (MHZU 23-0248). (**a–c**) Ascomata developing on *Wurfbainia villosa* stem. (**d,e**) Vertical section of an ascoma. (**f,g**) Section through the peridium. (**h,j,k**) Immature and mature asci. (**i**) Paraphyses. (**l–o**) Ascospores. (**p**) Ascospore in Indian ink. (**q**) Germinated ascospore. (**r**) Colony on PDA (front and below). Scale bars: (**d,h,i**) = 50 μm, **e** = 100 μm, (**f,g,j,k,q**) = 20 μm, (**l–p**) = 10 μm.

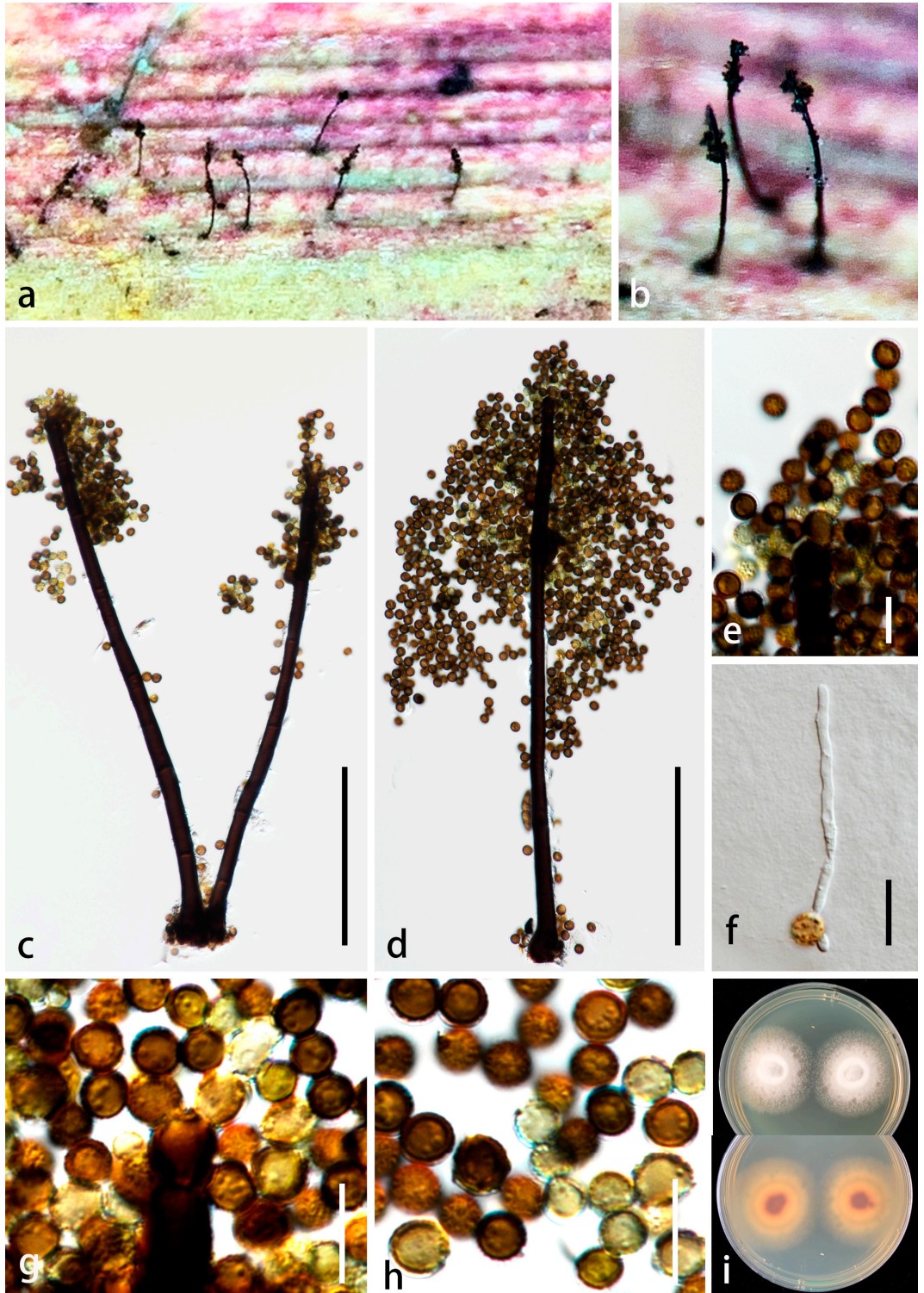

**Figure 4.** *Periconia wurfbainiae* (MHZU 23-0247, holotype). (**a**,**b**) Colonies on the host substrate. (**c**,**d**) Conidiophores with conidia. (**e**,**g**) Conidiogenous cell with conidia. (**f**) Germinated conidium. (**h**) Conidia. (**i**) Colony on PDA (front and below). Scale bars: (**c**,**d**) = 100 μm, (**e**–**h**) = 10 μm.

Material examined. China, Guangdong Province, Yangchun City, Heshui Town (22.32718988 N, 111.89251818 E), on dead stems of *Wurfbainia villosa*, 10 April 2022, C.F. Liao & Y.H. Yang, YAM28 (MHZU 23-0247, holotype); ex-type culture, ZHKUCC 23-0999; ibid. YAM29 (MHZU 23-0248), living culture ZHKUCC 23-1000.

Notes. In the phylogenetic analyses, *P. wurfbainiae* formed a distinct lineage from other *Periconia* species with 62% ML and 0.94 BYPP (Figure 1). *Periconia wurfbainiae* differs from its phylogenetically related species *P. variicolor* by having longer conidiophores (225–315 × 8–14 μm vs. 20–271 × 3–8 μm) and smaller conidia (3.5–7 μm vs. 7.5–9.5 μm). The presence of conidiophores with a stipe and swollen apex is absent in *P. wurfbainiae*, whereas they are observed in *P. variicolor* [45]. Moreover, the conidiophores of *P. wurfbainiae* are macronematous, dark brown to black, and smooth to slightly rough-walled. In contrast, *P. variicolor* has macronematous and micronematous conidiophores, initially subhyaline, becoming pale to dark brown at maturity, and smooth-walled [46]. Based on the distinct morphological characteristics and phylogenetic support, we introduce *P. wurfbainiae* as a new species.

*Periconia yangjiangensis* C.F. Liao, Doilom & K.D. Hyde, sp. nov.

Index Fungorum number: IF851410; Facesoffungi number: FoF 15278; Figure 5.

Etymology. In reference to the collection location, Yangjiang City, Guangdong Province of China.

Holotype. MHZU 23-0246

Description. Saprobic on dead stems of *Wurfbainia villosa*. Sexual morph: undetermined. Asexual morph: colonies superficial on the host substrate, hairy, gregarious, dark brown. Conidiophores 275–420 × 5–15 μm ($\bar{x}$ = 355 × 10 μm, *n* = 20), macronematous, mononematous, erect, subcylindrical, tapering towards the apex, with robust base, straight or slightly flexuous, dark brown to black in the lower portion, brown in the upper portion, with 1–4 times of enteroblastic percurrent proliferating, forming 1–4 swollen, pyriform or dumbbell-shaped, dark brown or paler cells from a collar cell, unbranched, sparsely septate, smooth-walled, thick-walled. Conidiogenous cells 15–25 × 8–10 μm ($\bar{x}$ = 18 × 9 μm, *n* = 10), holoblastic, poly-blastic, integrated, determinate, terminal, oblong, with rounded apex, wider than conidiophores, brown, smooth-walled, thin-walled. Conidia 6–10 μm diam. ($\bar{x}$ = 7.5 μm, *n* = 30), globose, catenate, mostly with three conidia in a chain, aseptate, initially hyaline, becoming golden brown at maturity, thick-walled, verruculose.

Culture characteristics. Conidia germinating on PDA within 24h, germ tube produced from both ends. Colonies on PDA reaching 35–45 diam., after two weeks at room temperature (25 ± 2 °C), medium dense, center raised, irregular, floccose to fluffy, velvety, lobate edge, golden brown at the center, pale brown at the margin from above and reverse.

Material examined. China, Guangdong Province, Yangchun City, Heshui Town (22.32718988 N, 111.89251818 E), on dead stems of *Wurfbainia villosa* (Zingiberaceae), 10 April 2022, C.F. Liao & Y.H. Yang, YAM04 (MHZU 23-0246, holotype); ex-type culture, ZHKUCC 23-0997; ibid., living culture ZHKUCC 23- 23-0998.

Notes. *Periconia yangjiangensis* resembles *P. pseudobyssoides* in the morphologies of conidiophores and conidia on the natural substrate [17,19]. However, conidia of *P. yangjiangensis* are initially hyaline, becoming golden brown at maturity, and verruculose but they are golden yellow to golden brown or reddish brown, spherical, verruculose when young, mature conidia with specific ornamentation consisting of irregular lobate crests in *P. pseudobyssoides* [17,19]. The morphological characteristics of *P. yangjiangensis* were also compared with those of *P. endophytica* and *P. wurfbainiae*, which were isolated from the same host plant, *Wurfbainia villosa*. The conidiophores of *P. yangjiangensis* are macronematous, dark brown to black, and unbranched, while they are micro and semimacronematous, hyaline, and branched in *P. endophytica*. The conidia of *P. yangjiangensis* are golden brown with typically three conidia found in a chain while they are pale brown to golden brown and typically form a long chain consisting of 4 to 10 conidia in *P. endophytica*. However, morphological differences could occur as *P. yangjiangensis* were directly observed on a dead host plant while *P. endophytica* were observed in PDA. *Periconia yangjiangensis* differs from *P. wurfbainiae*, in having conidiophores mostly 1–4 times of enteroblastic percurrent proliferating, forming 1–4 swollen, dark brown to black, paler towards the apex, terminal conidiogenous cells, with mostly three conidia in a chain. In contrast, *P. wurfbainiae* has conidiophores that are not swollen at the terminal cells,

dark brown to black, with both terminal and intercalary conidiogenous cells, usually 6–8 conidia in a chain. In addition, *P. yangjiangensis* can be distinguished from *P. wurfbainiae* by its longer conidiophores (275–420 × 5–15 µm vs. 225–315 × 8–14 µm) and larger conidiogenous cells (15–25 × 8–10 µm vs. 7–11 × 5–9 µm). In the phylogenetic analyses, *P. yangjiangensis* formed a distinct branch separated from other *Periconia* species with 95% ML, and 1.00 BYPP (Figure 1) indicating that *P. yangjiangensis* is distinct from other *Periconia* species. Based on morphological and phylogenetic support, we introduce *P. yangjiangensis* as a new species.

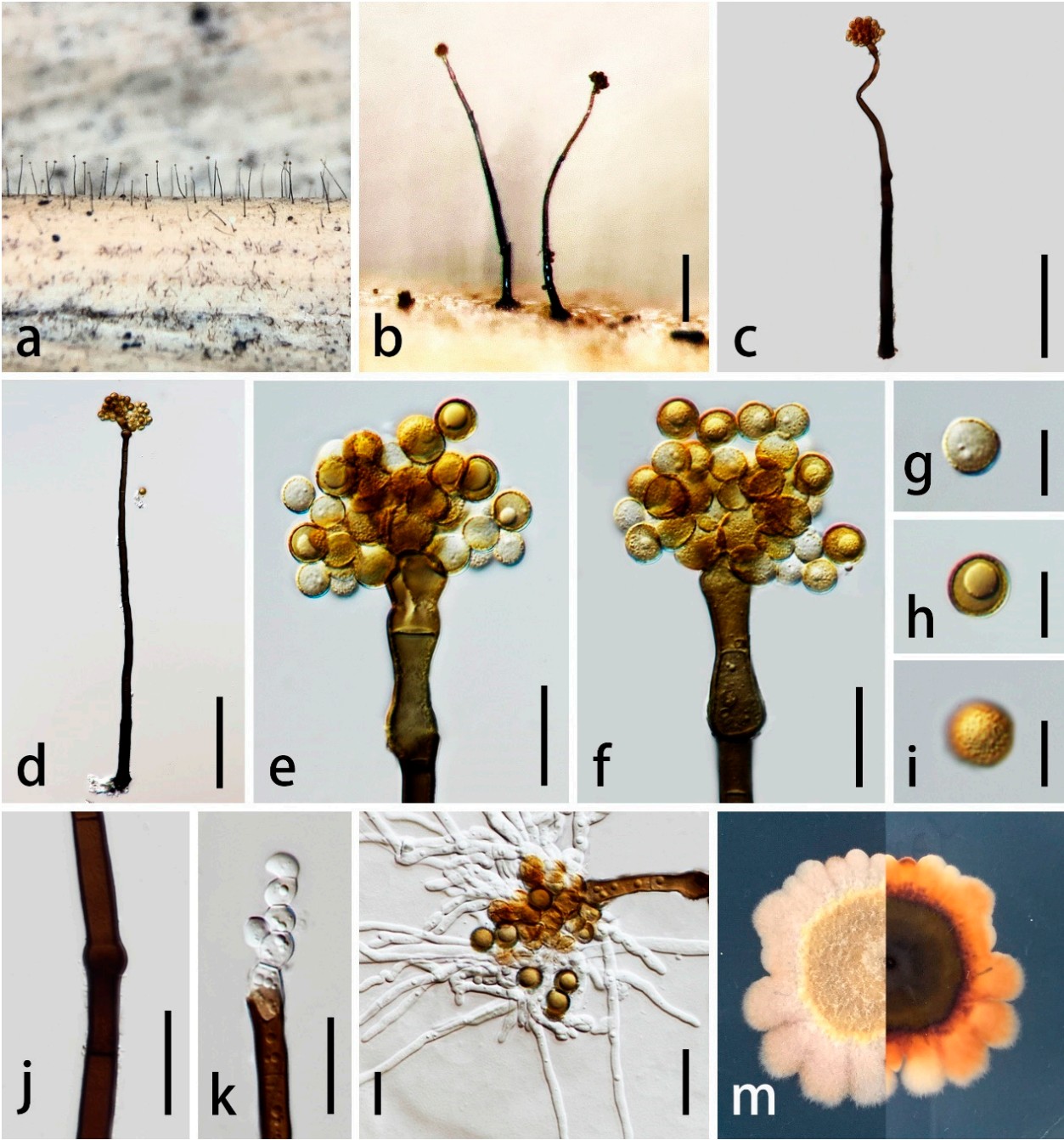

**Figure 5.** *Periconia yangjiangensis* (MHZU 23-0246, holotype). (**a,b**) Colonies on host substrate. (**c,d**) Conidiophore with conidia. (**e,f**) Swollen conidiophore, conidiogenous cell with conidia. (**g–i**) Conidia. (**j**) Swollen conidiophore. (**k**) Germinated conidiophore. (**l**) Germinated conidia. (**m**) Colony on PDA (front and below). Scale bars: 1 (**b–d**) = 100 µm, (**e,f,j–l**) = 20 µm, (**g–i**) = 10 µm.

**Table 2.** Synopsis of asexual morphological characteristics of *Periconia endophytica*, *P. yangjiangensis*, and *P. wurfbainiae* and its closely related species.

| Taxa | Conidiophores | Conidiogenous Cells | Conidia | Reference |
|---|---|---|---|---|
| *Periconia caespitosa* | Up to 500 μm long, 5–6 μm wide at the base, macronematous, mononematous, septate, unbranched or rarely branched, caespitose, straight to flexuous, setiform and sometimes uncinated at the tip, pale brown at the base, brown towards the apex, minutely roughened at the base and at the apex, as well as in the areas nearest of conidiogenous cells, otherwise smooth | (6–)7.5–9 × 6.5–7.5 μm, poly-blastic, pale brown, finely roughened, intercalary and terminal, globose, subglobose or obpyriform | (6–)6.5–9 μm, globose, aseptate, reddish brown, thick-walled, dry, solitary or in short basipetal chains of 2–4 conidia | Crous et al. [46] |
| *P. cambrensis* | 150–300 × 3–5 μm, erect, straight or tapering, pale brown, 5–9-septate, cylindrical, bearing a few short branches | 6–7 × 5–6 μm, ovate, hyaline or pale brown, smooth, bearing single or short-chained conidia | 5–8 μm diam., spherical, brown | Ellis [47] |
| *P. endophytica* | 15–110 × 2.5–5.5 μm (x̄ = 45 × 4 μm, n = 30), micro- to semi-macronematous, mononematous, erect, flexuous or straight, branched, sparsely septate, slightly constricted at septa, rough-walled, verruculose, thin-walled, hyaline, pale brown to yellowish brown in the upper portion | 5–20 × 2.5–5.5 μm (x̄ = 9 × 4 μm, n = 30), holoblastic, mono- and poly-blastic, integrated, determinate, terminal, subcylindrical, obovoid, ellipsoidal with truncate ends, pale brown to yellowish brown, verruculose | 4–6 × 3–5 μm (x̄ = 5 × 4 μm, n = 30), globose to subglobose, aseptate, catenate, with 4–10 conidia in a chain, hyaline when young, becoming pale brown, yellowish brown and reddish brown when mature, verruculose | This study |
| *P. epilithographicola* | 251.6–270 × 3.6–6.1 μm, macronematous, with creeping hyphae forming stipes straight, branched singly near the base, seven or more septate, grayish to black | (5.1–) 7 × 10 (−11.5) μm, holoblastic, sub-globose to ellipsoid, finely roughened, yellowish to brown | (7.8−) 9.2 (−10.7), globose, golden to brown, echinulated, catenated, sometimes forming long chains | Coronado-Ruiz et al. [48] |
| *P. neobrittanica* | 100–300 × 10–17 μm, solitary, or in clusters of 2–3, arising from a brown stroma, subcylindrical, straight to flexuous, unbranched, dark brown, smooth, thick-walled, base swollen, 15–25 μm diam. | 10–15 μm long, terminal and intercalary, occurring in an apical chain on primary, or directly on conidiophore | (6–)8–10(–12) μm diam., aseptate, spherical, pale to medium brown, with delicate spines, occurring in branched chains | Crous et al. [49] |
| *P. pseudobyssoides* | 23–25 μm wide at the base, up to 300–600 μm long, macronematous, arising usually singly, but occasionally 2 or 3(−4) together on stromata, often verruculose, with brown to reddish brown conidial heads | 10–12 × 6–7.5 μm, discrete, determinate, usually monoblastic, sometimes poly-blastic, ellipsoidal, ovoid to clavate, pale brown to reddish brown, verruculose, arising directly from the swollen mid-brown apical cell cut off by a septum from the stipe apex | (12–)15–17(−20) μm diam., spherical, golden yellow to golden brown or reddish brown, verruculose when young, mature conidia with specific ornamentation consisting of irregular lobate crests, born singly or in acropetal chains of 2–4 conidia | Markovskaja and Kačergius [19] |
| *P. variicolor* | 20–271 × 3–8 μm, simple, macronematous and micronematous, unbranched, mononematous, initially subhyaline, becoming pale to dark brown with age, smooth, occasional rigid hyphae formed but typical conidiophores with a stipe and swollen apex and base lacking | Discrete, determinate, terminal or lateral on the conidiophore, subglobose, ovoid to clavate, often poly-blastic. Terminal conidiogenous cells often form branched, acropetal chains that produce dense clusters of conidia | 7.5–9.5 μm, globose, one-celled, dark brown and verruculose at maturity | Cantrell et al. [45] |

**Table 2.** *Cont.*

| Taxa | Conidiophores | Conidiogenous Cells | Conidia | Reference |
|---|---|---|---|---|
| *P. wurfbainiae* | 225–315 × 8–14 μm ($\overline{x}$ = 265 × 10 μm, *n* = 20), macronematous, mononematous, sometimes 1–2 cluster together on the stroma, erect, subcylindrical, with robust and swollen base, straight or slightly curved, 5–7-septate, black, unbranched, smooth-walled to slightly rough | 7–11 × 5–9 μm ($\overline{x}$ = 8 × 6 μm, *n* = 10), holoblastic, poly-blastic, with several prominent conidiogenous loci at the apex, integrated, determinate, terminal and intercalary, reddish brown to black, subcylindrical, verruculose | 3.5–7 μm diam. ($\overline{x}$ = 5 μm, *n* = 15), globose, aseptate, catenate, usually 6–8 conidia in a chain, initially hyaline, becoming yellowish brown and greenish brown, and reddish brown to dark brown at maturity, thick-walled and verruculose | This study |
| *P. yangjiangensis* | 275–420 × 5–15 μm ($\overline{x}$ = 355 × 10 μm, *n* = 20), macronematous, mononematous, erect, subcylindrical, tapering towards the apex, with robust base, straight or slightly flexuous, dark brown to black in the lower portion, brown in the upper portion, with 1–4 times of enteroblastic percurrent proliferating, forming 1–4 swollen, pyriform or dumbbell-shaped, dark brown or paler cells from a collar cell, unbranched, sparsely septate, smooth-walled, thick-walled | 15–25 × 8–10 μm ($\overline{x}$ = 18 × 9 μm, *n* = 10), holoblastic, poly-blastic, integrated, determinate, terminal, oblong, with rounded apex, wider than conidiophores, brown, smooth-walled, thin-walled | 6–10 μm diam. ($\overline{x}$ = 7.5 μm, *n* = 30), globose, catenate, mostly with 3 conidia in a chain, aseptate, initially hyaline, becoming golden brown at maturity, thick-walled, verruculose | This study |

## 4. Discussion

Periconiaceae was resurrected by Tanaka et al. [11] and placed in Pleosporales based on morphological and phylogenetic evidence. This family comprises the genera *Bambusistroma*, *Flavomyces*, *Noosia*, and *Periconia* [11,50,51]. However, Yang et al. [16] re-evaluated the taxonomic status and synonymized *Bambusistroma* and *Noosia* under *Periconia* based on morphological characters of sexual morphs and multiloci phylogenetic analyses. *Flavomyces* was introduced by Knapp et al. [50] to accommodate the dark septate root endophyte, *F. fulophazii*, based on culture characteristics and phylogenetic analyses. Based on the phylogenetic analyses of combined ITS, LSU, SSU, and *tef1-α* sequence data in this study, *F. fulophazii* grouped within *Periconia* clade and formed a distinct lineage with *Periconia endophytica*, *P. yangjiangensis*, and *P. wurfbainiae* (Figure 1). We suggest that *F. fulophazii* should be synonymized with *Periconia*. However, the morphologies of *F. fulophazii* have not been described by Knapp et al. [51]. Therefore, more supporting evidence is required to validate the taxonomic placement of *F. fulophazii*.

*Sporidesmium tengii* (HKUCC 10837), which previously clustered closely with *P. didymosporium* in Yang et al. [16], groups within *Periconia* but forms a cluster with *P. cynodontis* (CGMCC 3.23927) in the current study with a 99% ML and 1.00 BYPP (Figure 1). This observed difference in the positioning could be attributed to differences in taxon sampling between the two studies. Unfortunately, no sequence data are available for the type of *S. tengii* [52]. Furthermore, the type of *S. tengii* described in Wu and Zhang [52] has macronematous, mononematous, erect, brown to dark brown conidiophores with terminal, brown, cylindrical, smooth-walled conidiogenous cells, which align with the generic characters of *Periconia*. However, the conidia of *S. tengii* differ from the generic characters of *Periconia* (globose to subglobose, aseptate), as they are acrogenous, solitary, obclavate, and 8-septate [52]. Otherwise, the strain of *S. tengii* (HKUCC 10837) that is clustered within *Periconia* only has the LSU sequence data [53] and its morphology has not been described in Shenoy et al. [53]. Given the contradictory evidence from different studies regarding the placement of *S. tengii* (HKUCC 10837), the taxonomic placement needs to be re-evaluated. Since *S. tengii* (HKUCC 10837) is only supported by LSU sequence data and lacks morphological support, it becomes challenging to transfer *S. tengii* (HKUCC 10837) to *Periconia*.

Therefore, morphological reexamination of the holotype of *S. tengii*, molecular data (ITS, SSU, and *tef1-α*) from an epi-type along with additional fresh collections are required to confirm the taxonomic placement.

The phylogenetic relationships among *Periconia* species were analyzed based on the combined ITS, LSU, SSU, and *tef1-α* sequence data, which supported the recognition of *P. endophytica*, *P. yangjiangensis*, and *P. wurfbainiae* as new species (Figure 1). Morphologically, *P. endophytica* resembles *P. cambrensis* in having conidia that are produced on conidiogenous cells and branches along the sides of the conidiophores [45]. However, the conidiophores of *P. endophytica* are shorter (15–110 μm) compared to those of *P. cambrensis* (up to 300 μm), while its conidia have a smaller diameter (4–6 μm) than those of *P. cambrensis* (5–8 μm) [47]. The sexual morph of *P. wurfbainiae* shares similar morphological characteristics with the sexual morphs of other *Periconia* species (e.g., *P. homothallica* and *P. pseudodigitata* in having immersed, globose ascomata with a papilla, cylindrical asci with a pedicel, and an ocular chamber, broadly fusiform, hyaline, ascospores, with nearly median septum, surrounded by a mucilaginous sheath [11]. However, *P. wurfbainiae* has larger ascomata (220–300 × 150–220 μm vs. 140–190 × 160–180 μm) and narrower asci (20–26 × 5–8 μm vs. 22–31 × 7–10 μm) compared to those of *P. homothallica*. *Periconia wurfbainiae* has larger ascomata (220–300 × 150–220 μm vs. 160–200 × 130–250 μm) with thicker peridium (13–30 μm vs. 6–13 μm) than those of *P. pseudodigitata* [11]. The morphological characteristics of the novel species can be distinguished from closely related taxa by examining the size and shape of their conidiophores, conidiogenous cells, and conidia, as detailed in the notes and Tables 2 and 3. To date, *Periconia* has 161 epithets listed in Species Fungorum (https://www.speciesfungorum.org/names/names.asp, accessed on 30 December 2023). Among these, approximately 133 have been accepted based on morphological characters, and 46 species, including three new species in this study, have molecular data. However, most *Periconia* species only have ITS and LSU sequence data. Therefore, it is recommended to incorporate additional loci (e.g., SSU and *tef1-α*) for accurate species identification. Also, more fresh collections are needed to reconfirm the taxonomic placements for the doubtful taxa mentioned before.

**Table 3.** Synopsis of sexual morphological characteristics of *P. wurfbainiae* and its closely related species.

| Taxa | Ascomata | Peridium | Asci | Ascospores | |
|---|---|---|---|---|---|
| *Periconia homothallica* | 140–190 μm high, 160–180 μm diam., scattered, immersed to erumpent, globose, with an ostiole | Longitudinal section uniformly 11–15 μm thick, composed of 4–6 layers of polygonal, thin-walled, 3–15 × 2–5 μm, pale brown cells | 85–119.5 × 13–17.5 μm ($\bar{x}$ 96.5 × 15.3 μm, *n* = 20), fissitunicate, cylindrical to lageniform, with a shallow ocular chamber, short-stalked (3.5–6 μm long), with 8 biseriate ascospores | 22–31 × 7–10 μm ($\bar{x}$ 26.3 × 8.7 μm, *n* = 60), l/w 2.6–3.7 ($\bar{x}$ 3.0, *n* = 60), broadly fusiform, with a nearly median septum (0.48–0.53; $\bar{x}$ 0.51, *n* = 38), hyaline, smooth, with an entire sheath; sheath gelatinous, up to 10 μm wide when fresh, later 1–2 μm wide | Tanaka et al. [11] |
| *P. pseudodigitata* | 160–200 μm high, 130–250 μm diam., numerous, scattered or 2–3 grouped, immersed to erumpent, globose | Longitudinal section 8–13 μm thick at side, 5–8 μm thick at the base, composed of 3–5 layers of thin-walled, 6–13 × 2–5 μm, brown cells | 70–110 × 10.5–15.5 μm ($\bar{x}$ 88.4 × 12.2 μm, *n* = 33), fissitunicate, cylindrical, rounded at the apex and with an apical chamber, short-stalked (5–15 μm long), with 8 irregularly biseriate ascospores | 19.5–27(–32) × 5–7 μm ($\bar{x}$ 22.5 × 6.1 μm, *n* = 134), l/w 2.9–4.5 ($\bar{x}$ 3.7, *n* = 134), broadly fusiform with rounded ends, straight or slightly curved, with almost median septum (0.48–0.55, $\bar{x}$ 5.1, *n* = 36), slightly constricted at the septum, hyaline, with or without guttules, smooth, with an entire sheath; sheath gelatinous, 1–2 μm wide at side and 2–4 μm wide at both ends in fresh, becoming delimited sheath in dry condition; senescent spores brown, echinulate, 1-septate | Tanaka et al. [11] |

**Table 3.** *Cont.*

| Taxa | Ascomata | Peridium | Asci | Ascospores | |
|---|---|---|---|---|---|
| *P. wurfbainiae* | 220–300 × 150–220 μm, solitary to scattered, immersed to semi-immersed, pyriform, black, papillate stain the substrate purple | 13–30 μm thick ($\overline{x}$ = 19 μm, *n* = 20), composed of 6–12 layers, hyaline to brown cells of *textura angularis* | 65–100 × 9.5–15 μm ($\overline{x}$ = 81 × 12 μm, *n* = 20), 8-spored, bitunicate, cylindrical, with rounded apex and narrower, short pedicellate, with an obscured ocular chamber | 20–26 × 5–8 μm ($\overline{x}$ = 23.5 × 7 μm, *n* = 30), overlapping uniseriate, biseriate in the middle portion, fusiform, slightly curved, 1-septate, slightly constricted at the septum, narrowly rounded at both ends, hyaline, guttulate, smooth-walled, surrounded by a mucilaginous sheath | This study |

**Author Contributions:** Conceptualization, C.L. and M.D.; methodology, C.L. and M.D.; software, C.L. and M.D.; validation, K.W.T.C., M.D., Y.Y., K.D.H. and W.D.; formal analysis, K.W.T.C., K.D.H. and M.D.; investigation, C.L., M.D., K.W.T.C., K.D.H. and W.D.; resources, C.L. and Y.Y.; data curation, C.L.; writing—original draft preparation, C.L. and M.D.; writing—review and editing, K.W.T.C., M.D., K.D.H. and W.D.; visualization, C.L.; supervision, M.D., K.W.T.C., K.D.H. and W.D.; project administration, M.D.; funding acquisition, M.D. All authors have read and agreed to the published version of the manuscript.

**Funding:** This study was supported by the Science and Technology Bureau of Guangzhou City (grant numbers 2023A04J1425 and 2023A04J1426), and the Innovative Team Program of the Department of Education of Guangdong Province (2022KCXTD015 and 2022ZDJS020). K.D.H. acknowledges the support of the National Research Council of Thailand (NRCT) (grant no. N42A650547) entitled "Total fungal diversity in a given forest area with implications towards species numbers, chemical diversity and biotechnology".

**Institutional Review Board Statement:** Not applicable.

**Data Availability Statement:** All data can be found in the manuscript.

**Acknowledgments:** We would like to express our gratitude to Shaun Pennycook (Landcare Research, New Zealand) for his critical nomenclatural review. We would also like to thank Zhongkai University of Agriculture and Engineering and Mae Fah Luang University for providing research facilities.

**Conflicts of Interest:** The authors declare no conflicts of interest.

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
