# Peer review of "Three New Periconia Species Isolated from Wurfbainia villosa in Guangdong, China: A Discussion on the Doubtful Taxa Clustering in this Genus"

_diversity, doi:10.3390/d16030141_

Round 1

Reviewer 1 Report

Comments and Suggestions for Authors

The manuscript is an interesting taxonomic study, the results of which are based on morphological and multilocus phylogenetic analyses and describe three new Periconia species found on the medicinal herb Wurfbainia villosa (Zingiberaceae) in China. The authors showed that the new species belongs to the ascomycetes of the genus Periconia, family Periconiaceae in the Pleosporales, Dothideomycetes. Two species, P. endophytica and P. yangjiangensis, were described based on their asexual morphs, Periconia wurfbainiae was described based on both asexual and sexual morphs. Descriptions of the new species are sufficiently informative and well-illustrated. The phylogeny of Periconia genus was performed using maximum likelihood (ML) and Bayesian inference (BI analyses and the best ML tree is presented. I recommend this manuscript for publication as an original paper. However, I would suggest improving the species descriptions and discussion section. I have several comments and questions listed below.

Firstly, I suggest correcting the

Keywords: traditional medicinal plant; microfungi, Periconiaceae; Ascomycota Zingiberaceae; multi-locus phylogeny; taxonomy

                             Introduction

Line 45. Please clarify taxonomic position of Periconia among Ascomycota. „..........plant W. villosa, ascomycetous fungi resembling Periconia ............................

Line 53. Please reject the term ‚hyphomycetes‘ „.....  asexual morphs (hyphomycetes), .....“

Line 59 ‚........ conidia [11,15, 18,19,11,15,20,21].

Line 63  ‚....sheath [16,17,22,23,16,17].

Line 80-81.   Please correct the sentence (it is very unclear) „Therefore, exploring the microfungi belonging to Periconia would be more conducive to revealing its potential biological compound value. “

Line 92-93. „Endophytic fungi were isolated from the healthy leaves of Wurfbainia villosa collected in

Yongning Town, Yangchun City, Guangdong Province, China in?. Please specify the year in which the leaves were collected.

Line 100-101. “Saprobic fungi were collected from the dead stem of Wurfbainia villosa in Heshui

Town, Yangchun City, Guangdong Province, China in ? Please clarify the year.

Line 105. Please correct this sentence „Fungal structures (e.g., ascomata, hamathecium, asci, and ascospores?) were examined “. What about structures of asexual morphs?

Line 151. „ ...generated using maximum-likelihood (ML) and Bayesian Inference (BI) analyses performed in the

Results

Line 184-486. In the Phylogenetic tree I see three genera: Lentithecium, Massarina and Morosphaeria. In the text the sentence „The phylogenetic tree of the combined ITS (1–651 bp), LSU (652–1939), SSU (1940–3350), and tef1-α (3351–4273) sequence data comprises 78 taxa, including Morosphaeria ramunculicola (KH 220) and M. velatispora (KH 221) as the outgroup taxa. “Please explain why you have included Lentithecium and Massarina species in your analysis. In the Materials and methods section there is a sentence „A total of 78 sequences were used in the phylogenetic analyses (Table 1), with Morosphaeria ramunculicola (KH 220) and M. velatispora (KH 221) as the outgroup taxa. “ And nothing about Lentithecium and Massarina species, including the Intoduction section.

Line 277. Please clarify - Conidiogenous cells of Periconia wurfbainiae are smooth or verruculose?

Line 282. Why you noted that conidia of Periconia wurfbainiae  is ? thin-walled? rough-walled. In the pictures I see that conidia are clearly thick-walled and verrucose. Please correct.

Line 283-287. Culture Characteristics. Please clarify the characterization of the different morphs, because in the photos I see some differences in cultures from ascospores and from conidia.

 Line 288. Culture characteristics. Please clarify whether Periconia wurfbainiae on PDA produced a reproduction structure of asexual morph or fruit bodies of a sexual morph.

 Line 236, 287.  What does that mean?  „...not producing pigmentation on PDA.“ I suggest delete.

In the picture I see light pigmentation in some parts of conidiophores and conidia.

Line 339. Please revise the description of Periconia yangjiangensis conidia.  I see that conidia are clearly thick-walled and verruculose.

Line 341-342. „Culture characteristics. ? Ascospores germinating on PDA within 24h, germ tube.

produced from both ends "What does that mean? On line 328 in the description of Periconia yangjiangensis, you write –„ Saprobic on dead stems of Wurfbainia villosa. Sexual morph: Undetermined. If you did not find sexual morph, why you write ascospores germinating? I think must be conidia germinating....

Line 350-351. Please clarify and correct „Notes. Periconia yangjiangensis resembles. P. seudobyssoides in their conidiophores and..“ I think must be P. pseudobyssoides. And include citation of original description of P. pseudobyssoides [17, 19]. Please check the original P. pseudobyssoides holotype description (KC954161) [19].

Line 353. Please correct this sentence „...yellowish brown to golden brown, echinulate, or verruculose in P. seudobyssoides [17]. You must compare with original description [19]. In original description we found that P. pseudobyssoides [19] produse verruculose conidiophores and verrucose conidiogenous cells and verrucose conidia, at maturity conidia becoming golden brown to reddish brown (12-)15-17(-20) µm diam. with a large lobate verruca.

Line 380. Table 2. Synopsis of morphological characteristics of Periconia endophytica, P.

yangjiangensis, and P. wurfbainiae and its closely related species. Please clarify - morphologically similar and phylogenetically closely related species.

Why did you not include in this table synopses of all morphologically similar species (which are under discussion) - P. cambrensis, P. pseudobyssoides, P. homothallica and P. pseudodigitata? In my opinion, it is not correct to give a synopsis of only the Asexual morph. If you want to give a table with synopsis of morphologically and phylogenetically close species of Periconia, it is better to give a complete synopsis of Asexual and Sexual morphs of all described and compared species in this article.

Discussion

Line 418-421. „The morphological characteristics of novel species can be distinguished from closely related species by examining the size and shape of their conidiophores, conidiogenous cells, and conidia, as detailed in the notes and Table 2. “I believe that this is insufficient.  I suggest moving some of the text from the notes after the description of the new species (where you compare morphology of close species) to the discussion section.

Comments on the Quality of English Language

Line 80-81.   Please correct the sentence (it is very unclear) „Therefore, exploring the microfungi belonging to Periconia would be more conducive to revealing its potential biological compound value. “

Author Response

Dear editor and reviewer(s),

Thank you very much for your review regarding our manuscript entitled “Three new Periconia species isolated from Wurfbainia villosa in Guangdong, China with discussion on the doubtful taxa clustering in this genus”. We sincerely appreciate the valuable comments and corrections to improve the quality of our research. The manuscript has been revised and modified in accordance with the referees' critiques. The revised texts have been visually emphasized with highlight. We hope that the revised version of our manuscript meets the requirements for publication in the journal.

Chunfang Liao, Kevin D. Hyde, Kandawatte Wedaralalage Thilini Chethana, Wei Dong, Yunhui Yang, and Mingkwan Doilom*

Reviewer 2 Report

Comments and Suggestions for Authors

Review on manuscript entitled “Three new Periconia species isolated from Wurfbainia villosa in  Guangdong, China with discussion on the doubtful taxa clustering in Periconia.” by Liao et al., for Diversity.

In general, very well thought and done manuscript, excellent mycology (phylogeny, descriptive taxonomical – phenotypical parts, excellent illustrations), well done introduction and discussion parts.

I have only several comments:

Title .. it is “clustering in Periconia.” Should be changed as “ clustering in this genus.” As Periconia then would appear twice in the same sentence. Or something similar. Alternatively to skip  completely “with discussion on the doubtful taxa clustering … this information the readers interested on Periconia and related fungi in general would find anyway in the text.

if the host plant Wurfbainia villosa has also a trivial name  I would use that name as well, 

Abstract: Line 17 it is P. endophytica should be “Periconia endophytica”

Line 17 family names in kursiva (latin) too  Periconiaceae”

Results Taxonomy:

Line 242 Taxon “P. cambrensis” is not in the phylo tree nor in the table ?

Line 244 it is “ … have shorter length “ should be  “ … are shorter”

In Table 2 , round off 251. 6 …  x 3.6 – 6.1  (252,  3.5 (or 4.0)  – 6.0)

---end--

Author Response

(The authors gave the same response as above.)

Reviewer 3 Report

Comments and Suggestions for Authors

The manuscript “Three new Periconia species isolated from Wurfbainia villosa in Guangdong, China with discussion on the doubtful taxa clustering in Periconia” provides descriptions of fungi isolated from Wurfbainia villosa, which is potentially interesting to the readers. However, the followings should be checked:

1.     The authors mentioned obtaining ITS, LSU, SSU, and tef1-alpha in the Material and methods section, but only three of these components are listed in Table 1 for their collections.

2.     The phylogeny is construct based on concatenated sequences form the collections and those from previous studies. It is found that each species have different parts of sequences, can this accurately represent the phylogenetic relationships?

3.     In figure 3, Periconia wurfbainiae (MHZU 23-0248) is used but it is not the holotype. There is a lack of information of MHZU 23-0248 in this article. More importantly, it is written that the species is based on the Material examined holotype of MHZU 23-0247. Why MHZU 23-0247 is not used for the figure?

4.     It appears that the authors use only one collection to obtain two isolates to represent each species, is it sufficient and unbiased?

5.     The measurement of the conidia is inconsistent among the three fungi. Periconia endophytica has the length and width, while the other two provide only the diameters. Why does P. yangjiangensis has include only the diameter but has the description of subglobose?

Additional comments:

L62: smooth-walled] -> smooth-walled

L109: potato 200g/L, dextrose 15g/L, agar 15g/L: check the blank key

Page 6, the lower line: P. yangjiangensis ZHKUCC 23-0100, it should be 23-1000 based on the study.

L180: It should be written as "National Chiayi University Culture Collection, Taiwan."; otherwise, "National Chiayi University Culture Collection."

Page 8, figure1: what’s the meaning of the arrows is not indicated and is it necessary?

L239: It should be written as (22.256185 N, 111.609037 E); italicized Wurfbainia villosa.

Author Response

(The authors gave the same response as above.)

Round 2

Reviewer 3 Report

Comments and Suggestions for Authors

Manuscript improved. Previous comment, now L236: 22.256185 N, 111609037 should be written as 22.256185 N, 111.609037 E

Author Response

(The authors gave the same response as above.)
